# Antimicrobial Activity of L-Lysine and Poly-L-Lysine with Pulsed Electric Fields

**Jurgita Švedienė [1],\*** , **Vitalij Novickij [2]**, **Rokas Žalnėravičius [3]** , **Vita Raudonienė [1]**, **Svetlana Markovskaja [4]** , **Jurij Novickij [2]** and **Algimantas Paškevičius [1]**

[1] Laboratory of Biodeterioration Research, Nature Research Centre, 08412 Vilnius, Lithuania; vita.raudoniene@gamtc.lt (V.R.); algimantas.paskevicius@gamtc.lt (A.P.)
[2] Faculty of Electronics, Vilnius Gediminas Technical University, 08412 Vilnius, Lithuania; vitalij.novickij@vilniustech.lt (V.N.); jurij.novickij@vgtu.lt (J.N.)
[3] State Research Institute Center for Physical Sciences and Technology, 03227 Vilnius, Lithuania; rzalneravicius@gmail.com
[4] Laboratory of Mycology, Nature Research Centre, 08412 Vilnius, Lithuania; svetlana.markovskaja@gamtc.lt
[*] Correspondence: jurgita.svediene@gamtc.lt; Tel.: +370-60-787-984

**Abstract:** For the first time, the possibility to use L-lysine (Lys) and poly-L-lysine (PLL) as additives with pulsed electric fields (PEF) for antimicrobial treatment is reported. The antimicrobial efficacy of Lys and PLL for *Escherichia coli*, *Staphylococcus aureus*, *Trichophyton rubrum* and *Candida albicans* was determined. Inactivation of microorganisms was also studied by combining Lys and PLL with PEF of 15 and 30 kV/cm. For PEF treatment, pulses of 0.5, 1, 10 or 100 µs were applied in a sequence of 10 to 5000 at 1 kHz frequency. The obtained results showed that 100 µs pulses were the most effective in combination with Lys and PLL for all microorganisms. Equivalent energy PEF bursts with a shorter duration of the pulse were less effective independently on PEF amplitude. Additionally, various treatment susceptibility patterns of microorganisms were determined and reported. In this study, the Gram-negative *E. coli* was the most treatment-resistant microorganism. Nevertheless, inactivation rates exceeding 2 log viability reduction were achieved for all analyzed yeast, fungi, and bacteria. This methodology could be used for drug-resistant microorganism's new treatment development.

**Keywords:** electroporation; microbial inactivation; *Escherichia coli*; *Staphylococcus aureus*; *Trichophyton rubrum*; *Candida albicans*

## 1. Introduction

Intensive use of antibiotics, drugs and food antimicrobials affect the occurrence of resistant microbial strains and thus limit the effectiveness of chemical methods in medical or biotechnological applications [1,2]. The arising problem of multi-drug-resistant strains [3] is ranked as one of the main priorities to focus on according to World Health Organization [3,4]. One of the typical solutions to antimicrobial resistance is an application of antibiotic cocktails, which has proven to be effective in many cases [5,6]. However, it is only a matter of time before microorganisms adapt and develop new resistance [5]. In food processing, the typical strategy is to use thermal treatment [7], but, in this case, the quality of food, taste and texture can be affected [8]. Therefore, one of the solutions is to find a synergistic or purely physical method, which could show minimal negative effects and allow successful inactivation of pathogenic microorganisms both in biomedical and biotechnological applications.

In the past few years, the antimicrobial feasibility of ultrasound [9,10], photodynamic therapy [11,12], cold plasma [13,14] and pulsed electromagnetic fields [15,16] was intensively studied. So far, one of the most successful treatments proposes the use of pulsed electric fields (PEFs) where the working principle is based on the cell electroporation [17,18]. This could increase the cell membrane permeability to exogenous molecules when the

cells are exposed to high-intensity PEFs [19]. This methodology is very flexible in terms of parameters and there is a possibility to apply reversible [20] and irreversible permeabilization [21]. This already has found application in the food industry [22], clinics [23] and biotechnology [24]. Recently, potential application for wound decontamination was described [25–27]. Furthermore, electroporation is very useful when the targeted chemical compounds delivery is required. For example, in cancer treatment, electrochemotherapy is used for drug delivery, which significantly reduces the toxicity of the treatment [28]. In food processing, PEF-based delivery of nanoparticles [29], nisin [30] or other bioactive compounds [31] is also possible, which usually leads to non-thermal bacteria inactivation.

Recently, it was demonstrated that PEF can be used for antibiotics for bacteria sensitization [32–34]. However, it does not solve the resistance problem entirely. Therefore, to find new agents to treat bacterial or other infections is an important and challenging task. The possibility to use electroporation could improve the treatment. In this work, we studied the antimicrobial properties of L-lysine (Lys) and poly-L-lysine (PLL) by combining them with PEF. The L-lysine was selected because it contains positively charged cationic groups of amino acids, which are very effective in destroying the membranes of the bacteria, thereby killing the bacteria [35,36]. However, poly-L-lysine shows antibacterial properties, can resist in a harsh environment, induces no inflammation of the tissue [37] and can be toxic only in high doses [38]. In this study, for the first time, the possibility to use L-lysine and poly-L-lysine as additives for antimicrobial therapy when combing with PEF is reported. This new method could be potentially applied both in biomedical and food processing areas.

Nevertheless, it should be noted that PEF-based treatment depends on the pulse parameters [39]. In this study, typical amplitudes of 15 and 30 kV/cm, usually used in antimicrobial studies, were applied [26,34]. The effect of pulse durations in the range of 500 ns to 100 μs were also investigated. Electroporation is a polarization-based phenomenon [40,41], and this could be the diminished effect of sub-microsecond duration due to the cell polarization constant being comparable or higher than the pulse duration. Our study presents a multiparametric approach to define the PEF effects by combining L-lysine and poly-L-lysine.

## 2. Materials and Methods

### 2.1. Preparation of L-Lysine and Poly-L-Lysine

L-lysine (Lys, 98%) and poly-L-lysine (PLL, 0.1%, MW 150-300 kDa) were purchased from Sigma-Aldrich Chemical Co, Germany and used as received without further purification unless otherwise stated. The final concentration of Lys and PLL of 1 mg/mL were freshly prepared in distilled water (DI). The Milli-Q DI (18.2 MΩ·cm) flex system produced by ELGA Purelab was used for the preparation of all solutions.

### 2.2. Preparation of Cells

In the preparation of cells suspension *Trichophyton rubrum* (Castell.) Sabour (ATCC 28188), *Candida albicans* (C.P. Robin) Berkhout (CBS 2730), *Staphylococcus aureus* Rosenbach (ATTC 43300) and *Escherichia coli* (Migula) Castellani and Chalmers (ATCC 25922) were used. Microorganisms were stored at $-70\ ^\circ$C in the Laboratory of Biodeterioration Research of the Nature Research Centre, Vilnius. *T. rubrum* was subcultured on potato dextrose agar (PDA) (Liofilchem, Roseto degli Abruzzi (TE), Italy). The suspension was prepared from fresh, mature cultures (7 days) in 1M sorbitol (ROTH, Karlsruhe, Germany). The suspension was homogenized for 15 s with vortex and then filtered using the sterile filter (11 μm). *T. rubrum* suspension optical densities were adjusted to 0.15 and 0.17 with yields of $0.7 \times 10^6$ and $1.2 \times 10^6$ CFU/mL, respectively. *C. albicans* was grown on a yeast extract peptone dextrose (YEPD) (2% glucose, 2% peptone, 1% yeast extract, and 1.5% agar) at $30\ ^\circ$C for 48 h. The *C. albicans* suspension concentration of $10^9$ CFU/mL was prepared in 1M sorbitol. *S. aureus* and *E. coli* were subcultured on nutrient agar (NA) (Liofilchem,

Roseto degli Abruzzi (TE), Italy). The bacteria suspension concentration of $10^8$ CFU/mL was prepared in 1M sorbitol after 24 h growth at 37 °C.

### 2.3. Antimicrobial Activity of L-Lysine and Poly-L-Lysine

The disk diffusion test was performed using Mueller–Hinton agar for fungi supplemented with 2% glucose. For *S. aureus*, 2% ($w/v$) of NaCl was used. For microorganisms testing, the inoculum of approximately $10^5$ CFU/mL for fungi and $10^8$ CFU/mL for bacteria was prepared using a sterile salt solution of 0.9%. The sterile disk (diameter 6 mm) was placed onto the inoculated agar surface. The amount of 10 μL of PLL (1 mg/mL) and Lys (1 g/mL) were placed on disk. The plates were incubated at $35 \pm 2$ °C. The tests were repeated in triplicate.

The minimal inhibitory concentrations (MIC) of Lys and PLL for *E. coli*, *S. aureus*, *T. rubrum* and *C. albicans* were determined according to the broth microdilution procedure EUCAST E. Dis 5.1 [42].

### 2.4. Pulsed Power Setup

In this study, the 0–3 kV square wave (100 ns–1 ms) electric pulse generator was used (VGTU, Vilnius, Lithuania) [43]. As a load (BTX, Cuvette plusTM, Nr. 610, San Diego, CA, USA), the commercially available electroporation cuvette with a 1-mm gap between the electrodes was used. The repetition frequency of 1 kHz with different durations between 500 ns and 100 μs was used for setup generated single and bursts of 0–30 kV/cm electric field. The 500 ns pulses were delivered in bursts of 2000, 1 μs pulses in bursts of 1000, 10 μs pulses in bursts of 100 and 100 μs in bursts of 10 with identical energy density. To compare the results, the 2.5-fold higher energy bursts were also introduced using pulses of 5000, 2500, 250 and 25, respectively.

### 2.5. Electroporation and Viability Assay

For the electroporation procedure, a suspension of respective cells was mixed with Lys at a final concentration of 500 mg/mL and PLL of 0.5 μg/mL. For pulsing, 80 μL samples of suspensions were used. After electroporation, *T. rubrum* was plated on the PDA and CFUs were counted after 4 days of incubation at 30 °C. *C. albicans* was plated on the YEPD and CFUs were counted after 48 h of incubation at 30 °C. *E. coli* and *S. aureus* were plated on the Muller–Hinton agar (Oxoid, Hampshire, England), and CFUs were counted after 24 h of incubation at 37 °C. The samples of each suspension without any treatment were used as control. The tests were repeated in triplicate. The inactivation efficacy was reported as log reduction of $CFU_T/CFU_C$, where $CFU_T$—CFU of the treated sample, $CFU_C$—CFU of the untreated control.

### 2.6. Statistical Analysis

For treatment comparison, one-way analysis of variance (ANOVA; $p < 0.05$) was used. If ANOVA showed significant effects, multiple Tukey HSD tests were used ($p < 0.05$ was considered significant) to evaluate the difference.

## 3. Results

To test the effect of Lys and PLL on the growth of *E. coli*, *S. aureus*, *C. albicans* and *T. rubrum* microorganisms, the disk diffusion test was performed. First, the antimicrobial efficacy of Lys and PLL was determined using the same concentration of 1 mg/mL. Obtained results are presented in Figure 1. No inhibition zone diameters of L-lysine at 1 mg/mL concentration were observed on tested microorganisms. As it can be observed, the antimicrobial efficiency on tested microorganisms was stronger using PLL at a concentration of 1 mg/mL (Figure 1).

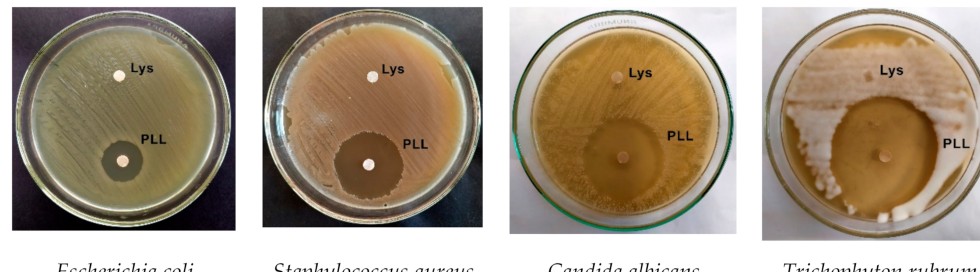

| *Escherichia coli* | *Staphylococcus aureus* | *Candida albicans* | *Trichophyton rubrum* |

**Figure 1.** The antimicrobial efficacy of L-lysine ((Lys) (disk 1 mg/mL)) and poly-L-lysine ((PLL) (disk 1 mg/mL)) on microorganisms.

Since microorganisms show different sensitivity, the minimal inhibitory concentration (MIC) was performed to find the right concentration for each substance. The effective concentrations are presented in Table 1.

**Table 1.** Minimal inhibitory concentration of L-lysine (Lys) and poly-L-lysine (PLL) on microorganisms.

| Microorganisms | Lys | PLL |
|:---:|:---:|:---:|
| | MIC | |
| *E. coli* | 500 mg/ml | 250 mg/ml |
| *S. aureus* | 125 mg/ml | 2.4 ng/ml |
| *C. albicans* | NI * | 1.5 ng/ml |
| *T. rubrum* | NI | 0.3 ng/ml |

* NI—no inhibition at 1g/mL concentration of L-lysine.

PLL exhibited very strong antimicrobial activity on all tested microorganisms. However, Lys had no effect on the growth of *C. albicans* and *T. rubrum*. According to these results, the selected concentration to use with PEF treatment for Lys was 500 mg/mL and for PLL 0.5 was μg/mL.

Since the highest activity was observed on *T. rubrum,* the first treatment using PEF and Lys/PLL was analyzed and results are presented in Figure 2. Indeed, all the data was normalized to non-treated control.

As can be seen in Figure 2A, both Lys and PLL can be effectively used with electroporation for inactivation of *T. rubrum*. PLL features better antimicrobial properties. The duration of the PEF pulses was varied, and different efficacies of the treatment were acquired. The energy of the bursts was the same independent on the pulse duration, however, the 500 ns and 1 μs treatments showed significantly weaker effects compared to the 10 and 100 μs bursts. When the number of pulses was increased (2.5-fold energy increase), the efficacy of the treatment was observed. However, the increase of PEF amplitude to 30 kV/cm significantly improved the inactivation efficacy (Figure 2B). The viability drop was more than 3 log reduction for the highest energy burst (100 μs × 25). Similar to the 15 kV/cm treatment, the 500 ns and 1 μs protocols were less effective. The differences between Lys and PLL were not as reflective as in 15 kV/cm (except the highest energy burst).

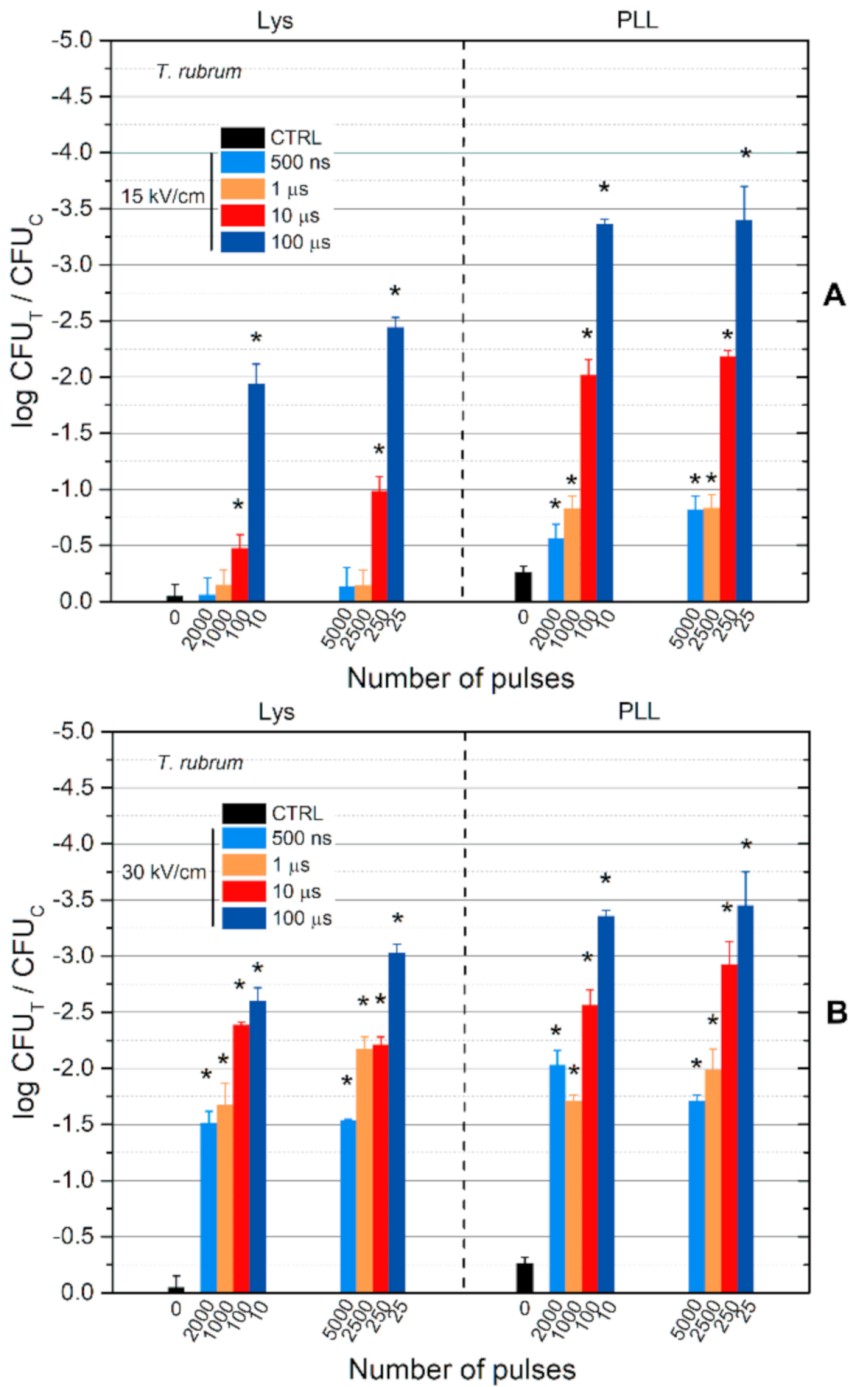

**Figure 2.** The antimicrobial efficacy of L-lysine and poly-L-lysine with pulsed electric field treatment against *T. rubrum*, where (**A**) 15 kV/cm treatment; (**B**) 30 kV/cm treatment. Asterisk (*) represents statistically significant ($p < 0.05$) difference versus Lys or PLL treatment, respectively. CTRL—samples with Lys or PLL without PEF treatment.

Further, the susceptibility of *C. albicans* to the treatment was analyzed. The results are summarized in Figure 3.

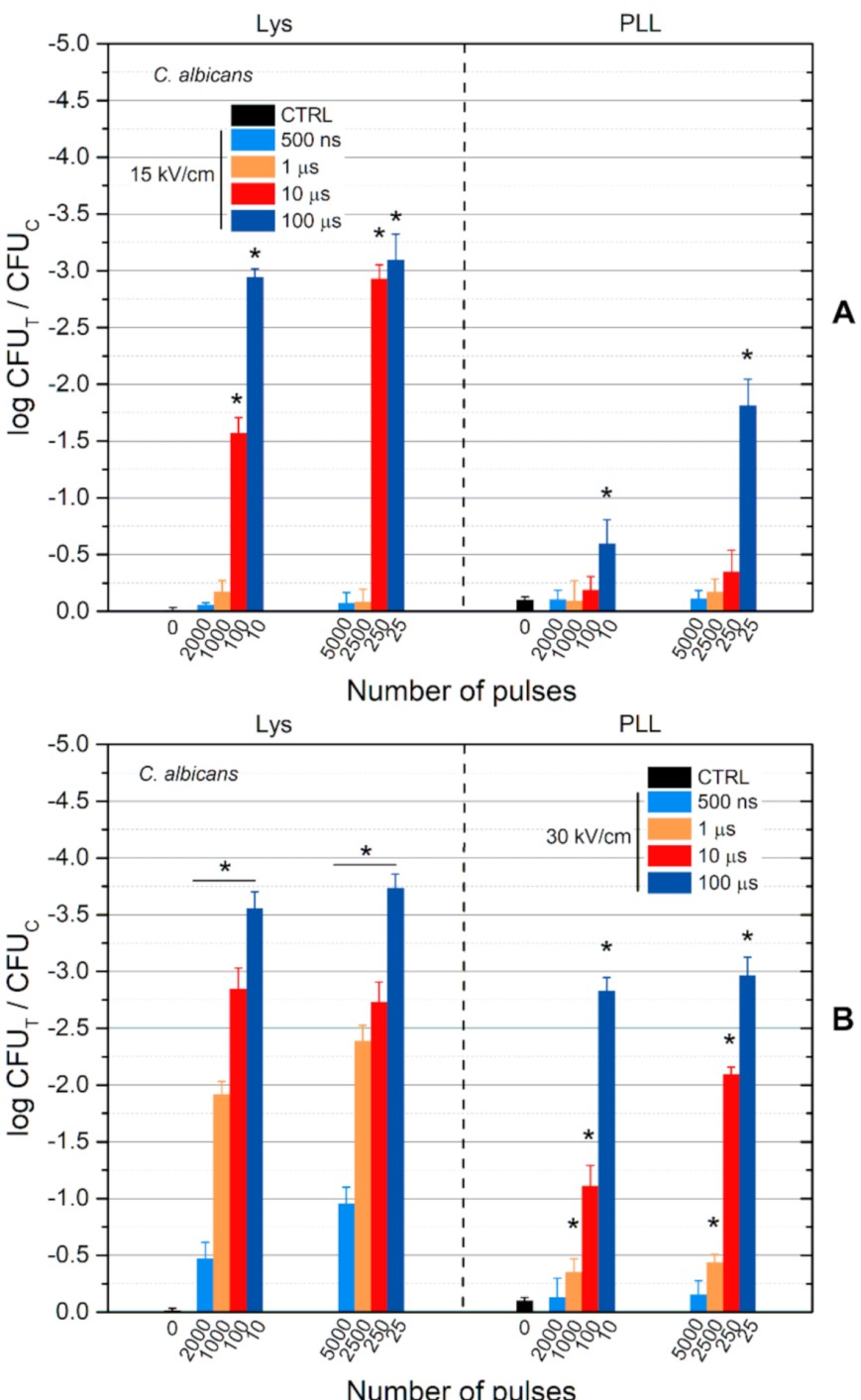

**Figure 3.** The antimicrobial efficacy of L-lysine and poly-L-lysine with pulsed electric field treatment against *C. albicans*, where (**A**) 15 kV/cm treatment; (**B**) 30 kV/cm treatment. Asterisk (*) represents statistically significant (*p* < 0.05) difference versus Lys or PLL treatment, respectively. CTRL—samples with Lys or PLL without PEF treatment.

As can be seen in Figure 3, a similar tendency was observed, i.e., the 500 ns and 1 μs treatments are the least effective. *C. albicans* demonstrated higher susceptibility to the combination of PEF + Lys compared to *T. rubrum*. Also, when combined with PEF, the PEF + Lys treatment was more effective than PEF + PLL treatment. In general, the response of *C. albicans* to PEF + PLL was significantly weaker (*p* < 0.05) versus *T. rubrum*.

Further, the inactivation of bacteria was also studied. The results for *S. aureus* are presented in Figure 4.

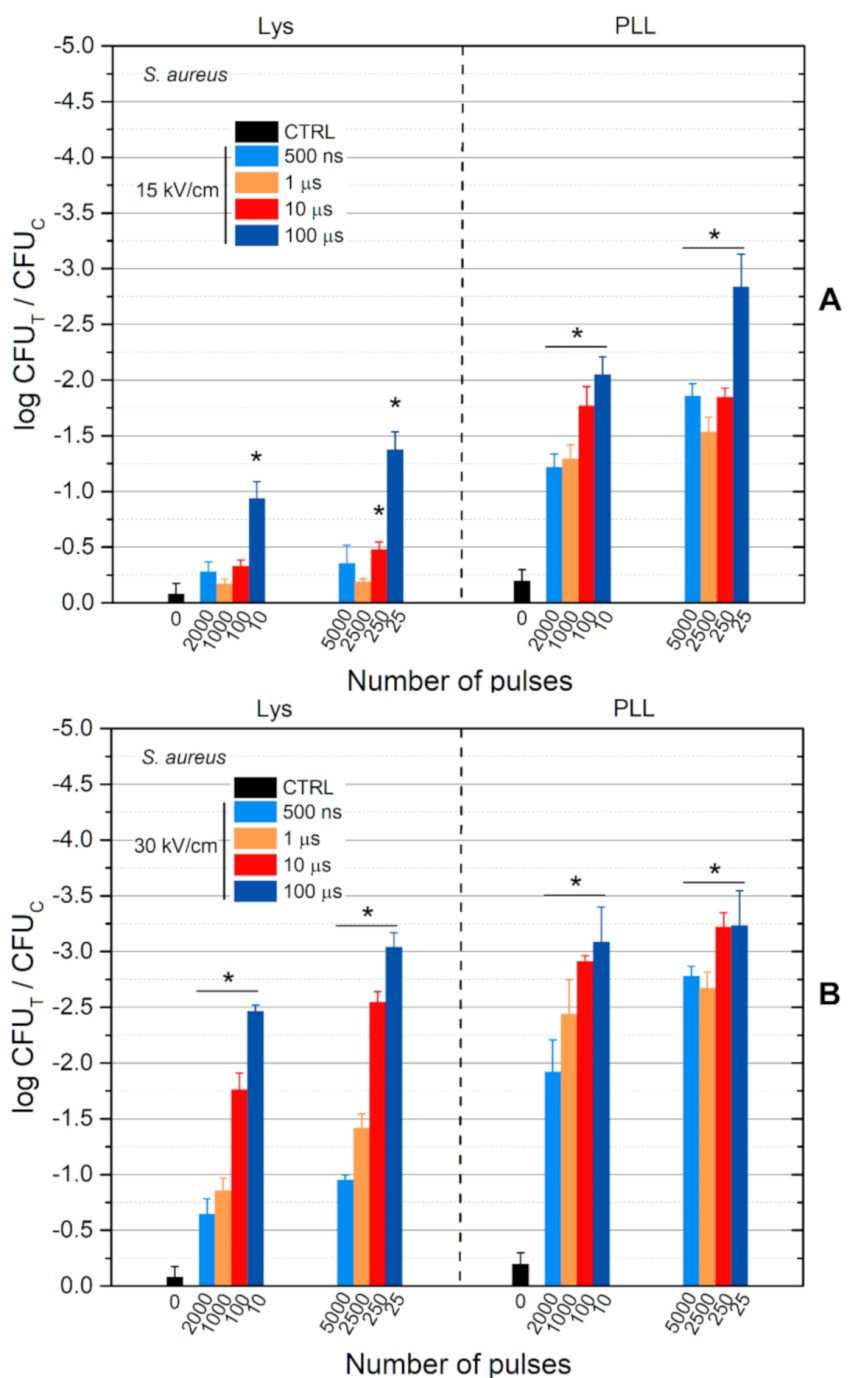

**Figure 4.** The antimicrobial efficacy of L-lysine and poly-L-lysine with pulsed electric field treatment against *S. aureus*, where (**A**) 15 kV/cm treatment; (**B**) 30 kV/cm treatment. Asterisk (*) represents statistically significant ($p < 0.05$) difference versus Lys or PLL treatment, respectively. CTRL—samples with Lys or PLL without PEF treatment.

It can be seen, that both Lys and PLL in combination with PEF are effective against Gram-positive bacteria *S. aureus*. Duration-wise the tendency observed with previous microorganisms persists. Similarly, the increase in the number of pulses (2.5-fold) did not improve the treatment efficacy significantly.

However, as was expected, an increase of the PEF amplitude to 30 kV/cm (Figure 4B) significantly improved the inactivation efficacy of shorter duration pulses (500 ns and 1 μs).

Lastly, the efficacy of the treatment on the Gram-negative bacteria *E. coli* was investigated. The results are presented in Figure 5. The antimicrobial response against *E. coli*

was the weakest compared to all the other microorganisms used in this study. The increase of the number of pulses was not effective; however, the treatment with 30 kV/cm field in combination with Lys and PLL improved the inactivation several-fold.

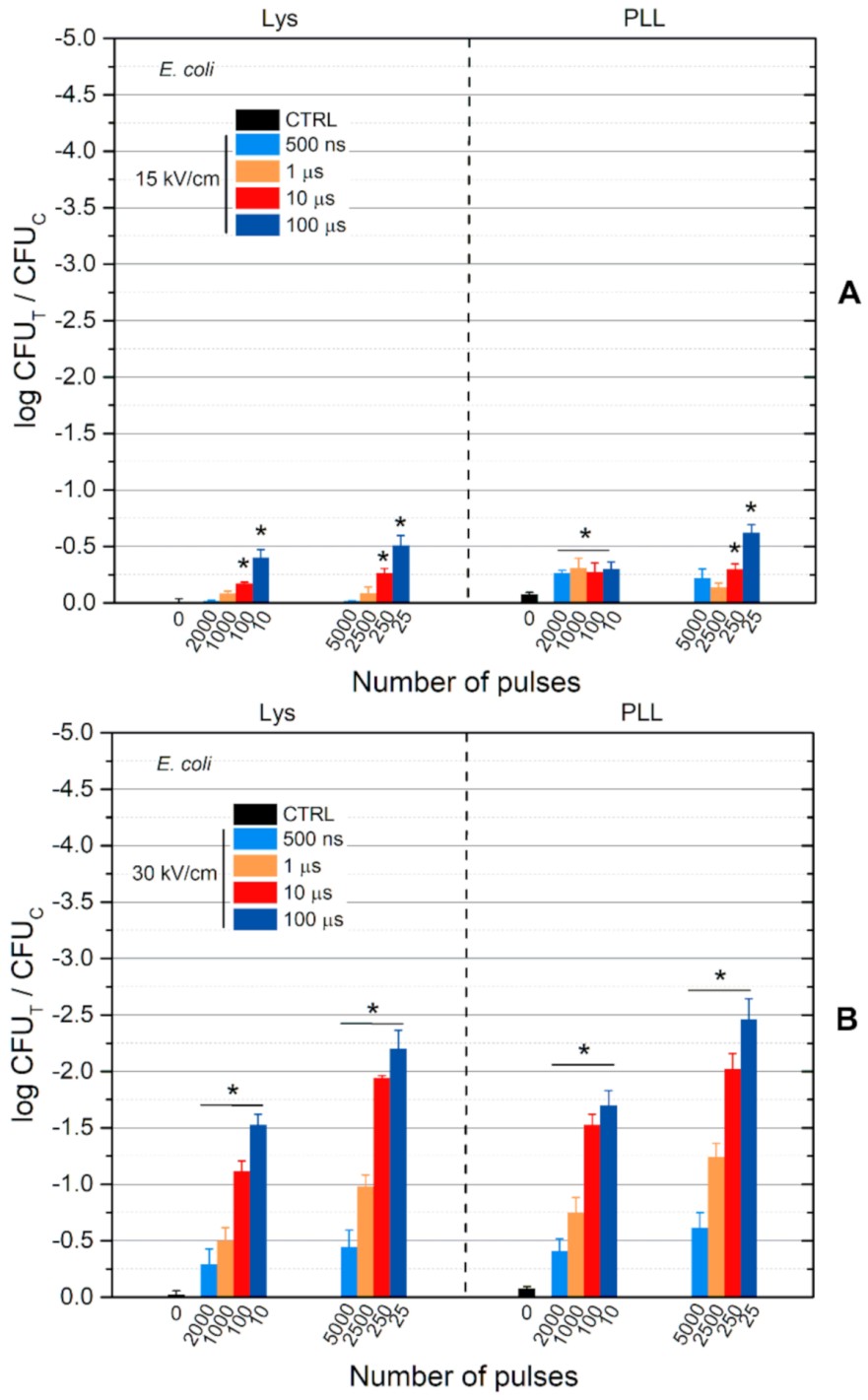

**Figure 5.** The antimicrobial efficacy of L-lysine and poly-L-lysine with pulsed electric field treatment against *E. coli*, where (**A**) 15 kV/cm treatment; (**B**) 30 kV/cm treatment. Asterisk (*) represents statistically significant ($p < 0.05$) difference versus Lys or PLL treatment, respectively. CTRL—samples with Lys or PLL without PEF treatment.

The scaling with the number of pulses follows a predictable manner, indicating that longer duration pulses are more efficient compared to the sub-microsecond procedure. An increase of the burst energy via an increase of the number of pulses (2.5-fold) positively

affects the treatment with 1 μs pulses and 30 kV/cm treatment (Figure 5B). The inactivation efficiency scales proportionally (more than 2-fold).

## 4. Discussion

Pulsed electric fields are commonly applied for microbial inactivation in food processing [31]. The use of PEF with antimicrobial components could lead to new developments for surface infections treatments [34,44]. In this work, we have investigated inactivation of microorganisms by combining Lys and PLL with PEF treatment. Four pathogenic microorganisms with different morphologies and membrane structures were used. Two fungal pathogens, *T. rubrum* and *C. albicans*, and two bacterial pathogens, *S. aureus* and *E. coli* were used. Generally, the susceptibility to electroporation depends on the biological object. Usually, bacteria show higher resistance to PEF when compared to other microorganisms like eukaryotes [45], requiring higher energy/PEF amplitude procedures [46]. Differences in susceptibility were also observed in our studies. The keratinophilic fungus known as dermatophyte *T. rubrum* was the most susceptible, followed by the Gram-positive bacteria *S. aureus* and human pathogenic yeast *C. albicans*. Though, the Gram-negative bacteria *E. coli* was the most resistant demanding the highest energy burst for successful inactivation. Nevertheless, the energy can be controlled by variation of many parameters such as PEF amplitude, number of pulses or the pulsed current. In our study, two amplitudes of 15 and 30 kV/cm were selected and the effect of the number of pulses and pulse duration on the treatment outcome was analyzed. According to our results, higher PEF amplitude resulted in better treatment efficiency using both Lys and PLL. However, to demonstrate the importance of polarization and the effects of electrotransfer on inactivation efficiency various pulse durations were analyzed. It was observed that sub-microsecond pulses show the weakest response of Lys and PLL treatment for studied microorganisms. However, longer pulse duration significantly improved the efficiency of the treatment for all pathogenic microorganisms when the same energy burst was used. The results showed that longer pulses ensure better electrotransfer of Lys and PLL, which can be explained by better pore stability [47] and higher electrophoretic force [48].

In our work, the susceptibility of the microorganisms to Lys and PLL was also studied showing that the Gram-negative bacteria *E. coli* was the most resistant in both cases. According to several studies [49,50], the mechanism of well-known ε-PLL and PLL were considered to be similar based on their similar chemical properties. Possibly, positively charged L-Lys amino acid and PLL could electrostatically interact with neutral or negatively charged microorganism membrane systems. Hyldgaard's group [50] observed that the cationic polypeptide could interact with negatively charged cell surface by ionic adsorption inducing the stripping of the lipopolysaccharide (LPS) layer, which could increase permeabilization of the outer membrane and lead to rapid cell death. Probably, the cationic amino groups of Lys and PLL interact with the outer membrane of Gram-negative bacteria containing hydrophilic and negatively charged LPS. However, the Gram-negative microorganism membrane is composed of two membranes and a thin layer of peptidoglycan between them, leading to a more resistant structure for the positively charged organic molecules diffusion. For the Gram-positive bacteria, the negative charges are supplied by carboxylate, phosphate or sulphate groups of anionic polymers called teichoic acids. Recently, Tan et al. [51] reported that peptidoglycan structure from the Gram-positive *S. aureus* can be affected by ε-PLL. This observation could indicate that Lys and PLL can induce damage through the peptidoglycan layer making the cell wall more fragile. PEF technology can be used synergistically with heat, antimicrobial agents, membrane filtration and ultraviolet radiation to increase the effectiveness of bacterial inactivation prolonging the period of consumption. PEF studies have a wide range of applications for microbial inactivation in liquid and semisolid products (milk, milk products, egg products, juice, etc.) [8,10,52]. It is known that PEF treatment at 35 kV/cm and 90 μs can decrease by 5.15 log numbers of *E. coli* colonies. Although, PEF treatment at 20 kV/cm with 60 pulses provided nearly 2 log reduction in viable cell counts of *S. aureus* [53,54]. Therefore, modulation

of the inactivation efficiency based on PEF parameters is possible and higher inactivation rates can be achieved. Also, in this study, we used Lys and PLL concentrations, which cause less than <0.25 log reduction in all microorganisms when applied without PEF. Therefore, for further studies higher concentrations of Lys and PLL can be used.

Generally, PLL has higher cationic behavior than Lys due to the higher number of repetitive L-Lys residues, which can vary from 1000 to 2000. This could be the reason for PLL's higher antimicrobial activity, which was also observed by combining with pulsed electric field treatment. Additionally, the use of PEF with PLL or Lys could induce generation of negative curvature and may simplify interactions at the surface of the membrane. During this process, cationic amino groups induce negative curvature wrapping of anionic membranes and lead to micellization/vesiculation, which disrupts membrane integrity causing the thinning of membranes.

## 5. Conclusions

In this work, we have investigated the feasibility of pulsed electric fields (15, 30 kV/cm) delivered in bursts of 10–5000 pulses for potentiation of the antimicrobial properties of Lys and PLL. Four pathogenic microorganisms with different morphologies and membrane structures were used as models. Two fungal pathogens, *T. rubrum* and *C. albicans,* and two bacterial pathogens, *S. aureus* and *E. coli,* were analyzed. The results showed that the combination of Lys and PLL with PEFs induces strong antimicrobial effects. However, the susceptibility to the treatment depends on pathogen type, applied electric field, pulse length and pulse number. This methodology could be used for drug-resistant microorganism's new treatment development.

**Author Contributions:** V.N., R.Ž. and A.P. conceived the experiments and methodology. V.N. and J.N. developed the experimental setup and applicators. J.Š., V.N., S.M. and V.R. performed microbiological experiments. V.N., R.Ž., J.Š. and J.N. interpreted the results. V.N., R.Ž. and J.Š. wrote the manuscript. All authors reviewed the manuscript and provided valuable comments on the work. All authors have read and agreed to the published version of the manuscript.

**Funding:** This research was partly funded by the European Social Fund under the No 09.3.3-LMT-K-712-19-0155 "Development of Competences of Scientists, other Researchers and Students through Practical Research Activities" measure.

**Institutional Review Board Statement:** Not applicable.

**Informed Consent Statement:** Not applicable.

**Data Availability Statement:** Data available from the corresponding author J.Š. on request.

**Conflicts of Interest:** The authors declare no conflict of interest.

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
