# Peer review of "Antimicrobial Activity of L-Lysine and Poly-L-Lysine with Pulsed Electric Fields"

_applsci, doi:10.3390/app11062708_

Round 1
Reviewer 1 Report
This study provides some new knowledge and opportunities for safer food processing and applicability in medicine and pharmacy. After reading the manuscript, I did not notice any significant errors, nor spelling and stylistic errors, English language and style are also fine. For my part, I would recommend accepting this manuscript in it's present form.
Author Response
Dear Keith Wang (Assistant Editor) and Reviewers of Applied Sciences,
Re: Revision of “Antimicrobial Activity of L-Lysine and Poly-L-Lysine with Pulsed Electric Fields” by Jurgita Svediene, Vitalij Novickij, Rokas Zalneravicius, Vita Raudoniene, Svetlana Markovskaja, Jurij Novickij, Algimantas Paskevicius (applsci-1127909)
Thank you very much for your kind reviews and comments regarding our manuscript (applsci-1127909) entitled above. Now we have carried out revisions according to your comments and hope this will be adequate for the acceptance of this manuscript. Details of corrections according to the comments are as follows;
|
Reviewer #1 |
|
|
General Comment |
This study provides some new knowledge and opportunities for safer food processing and applicability in medicine and pharmacy. After reading the manuscript, I did not notice any significant errors, nor spelling and stylistic errors, English language and style are also fine. For my part, I would recommend accepting this manuscript in it's present form. |
|
Response to General Comment |
Thank you very much for your comments. |
Reviewer 2 Report
Summary: The authors of this article investigate the antimicrobial efficacy of L-lysine (Lys) and poly-L-Lysine (PLL) in conjunction with pulsed electric fields (PEFs). The authors show that PEFs significantly increase the antimicrobial efficacy of Lys and PLL against two fungal and two bacterial pathogens. Longer pulses (100 us) with Lys and PLL are significantly more effective than shorter pulses (500ns). Higher applied electric fields (30 kV/cm vs 15 kV/cm) significantly increase the antimicrobial efficacy, while increasing the pulse number by 2.5 times does not yield a substantial change in efficacy. Overall, the results show that the combinatorial treatment of Lys and PLL with PEFs show antimicrobial effects that are dependent on pathogen type, applied electric field, pulse length, and pulse number.
Recommendation: I recommend this paper be accepted after major revisions. The study presents interesting results that add to the existing literature on PEFs for antimicrobial applications. However, several revisions/changes should be made to improve the manuscript before publication.
Overall Comments:
- While the authors clearly show reduced viability for Lys and PLL treatments in conjunction with PEFs, it is not clear how the viability is reduced by Lys and PLL treatment alone (at the concentration used in the PEF experiments). Was a “sham” experiment performed where pathogens were “treated” with no voltage? Perhaps, this condition represents the 100% viability on plots, but if so, it is not clear from the manuscript. Please clarify the discussion and figures accordingly to clearly show/describe the additive effects of the PEFs compared to Lys and PLL alone. In addition, it could be valuable to perform a “sham” experiment using just PEFs without Lys or PLL to show that the reduced viability is not just due to the PEF treatment.
- It would be valuable for the authors to add a Conclusion section to summarize the findings from this paper.
- Unfortunately, frequent grammatical mistakes and awkward sentence structures make the paper difficult to read and understand in places. I recommend a thorough revision of the paper for grammar and style. Removing grammatical and style issues will greatly enhance the impact and clarity of the publication.
Additional Comments:
Introduction/Methods
- Pg 2, line 91. YEPD should be defined.
- Section 2.3. “2.5-fold higher energy bursts” is rather confusing. It would be more helpful to clarify this is due to higher pulse numbers, such as by stating “2.5-fold more pulses (2.5-fold increase in energy delivered).”
Results
- Figure 1. It seems like part B of this figure has been omitted. Please add this to the Figure. It could be helpful to at label each image with the pathogen type. Furthermore, clarify that the upper disk is a control (I assume that is the case).
- The authors should enhance and revise Table 1. It would be valuable to include the zone of inhibition diameters in addition to the equivalent MICs. Furthermore, the units of the MIC values should be double-checked. For C. albicans and T. rubrum, MIC values for Lys are indicated as “>10 g/l.” Why is the “>” necessary? 10g/l is equivalent to 10mg/ml, so I’m confused why mg/ml are not used here as in the other rows. Please revise.
- The rationale for selecting the concentrations of Lys and PLL should be further explained. It is not clear why these values were chosen for use with the PEF treatments.
- Figure 2. The final bar in the plot (25 pulses at 100us duration) is not visible in either plot. Therefore, it is not clear what this value for this condition is. This is understandable for part A, assuming the viability is very low; however, in part B with the log scale, it is unclear whether the value is very close to 0.01 (lowest value on the graph), or if it is below this value. I suggest decreasing the lower bound of the y-axis to show this final bar.
- Figure 5B. A log scale for viability would better show the data for this plot.
- It would be helpful for the authors to include a table summarizing the results from Figures 2-5. In particular, it would be helpful for the authors to state the log reduction of pathogens for each treatment condition.
Discussion
- The results show Lys and PLL treatment in combination with PEFs can reduce pathogen viability by several log reductions (~1-4). Can you discuss if these reductions are sufficient for disinfection proposes for food/biotechnology applications? It would be good to compare these efficacies with other similar studies (either with Lys/PLL or PEFs with other agents). If these efficacies are not low enough for some applications, could higher voltages or longer pulses potentially increase the efficacy?

Author Response
Dear Keith Wang (Assistant Editor) and Reviewers of Applied Sciences,
Re: Revision of “Antimicrobial Activity of L-Lysine and Poly-L-Lysine with Pulsed Electric Fields” by Jurgita Svediene, Vitalij Novickij, Rokas Zalneravicius, Vita Raudoniene, Svetlana Markovskaja, Jurij Novickij, Algimantas Paskevicius (applsci-1127909)
Thank you very much for your kind reviews and comments regarding our manuscript (applsci-1127909) entitled above. Now we have carried out revisions according to your comments and hope this will be adequate for the acceptance of this manuscript. Details of corrections according to the comments are as follows;
|
Reviewer #2 |
|
|
Overall Comments |
While the authors clearly show reduced viability for Lys and PLL treatments in conjunction with PEFs, it is not clear how the viability is reduced by Lys and PLL treatment alone (at the concentration used in the PEF experiments). Was a “sham” experiment performed where pathogens were “treated” with no voltage? Perhaps, this condition represents the 100% viability on plots, but if so, it is not clear from the manuscript. Please clarify the discussion and figures accordingly to clearly show/describe the additive effects of the PEFs compared to Lys and PLL alone. In addition, it could be valuable to perform a “sham” experiment using just PEFs without Lys or PLL to show that the reduced viability is not just due to the PEF treatment. |
|
Response to General Comment |
Thank you. All the data was normalized to non-treated control. A respective comment was added to the text and material & methods. The graphs have been also revised. We have included Lys and PLL only efficacies too. We did not plan the PEF-only experiments initially since the novelty of PEF-only treatment is limited in this context. There are multiple contributions focusing the sole PEF effects on the described microorganisms in a very systemic manner. To ensure actuality and novelty in our work we focused only the synergistic approaches and highlighted mainly the differences between Lys and PLL, which were not covered before in scientific literature. At the current situation (strict COVID-19 quarantine is extended until May, most labs are closed) it’s not possible to perform additional experiments in a reasonable timeframe. However, to account for the remark, we have added additional relevant citations in the discussion, which cover the PEF-only effects on the described microorganisms. |
|
Overall Comments |
It would be valuable for the authors to add a Conclusion section to summarize the findings from this paper. |
|
Response to General Comment |
We have added a Conclusion section. |
|
Overall Comments |
Unfortunately, frequent grammatical mistakes and awkward sentence structures make the paper difficult to read and understand in places. I recommend a thorough revision of the paper for grammar and style. Removing grammatical and style issues will greatly enhance the impact and clarity of the publication. |
|
Response to General Comment |
We have revised the manuscript. |
|
Additional Comments |
Pg 2, line 91. YEPD should be defined. |
|
Response to Additional Comment |
Fixed. |
|
Additional Comments |
Section 2.3. “2.5-fold higher energy bursts” is rather confusing. It would be more helpful to clarify this is due to higher pulse numbers, such as by stating “2.5-fold more pulses (2.5-fold increase in energy delivered).” |
|
Response to Additional Comment |
We have improved the text in accordance with the recommendation. |
|
Additional Comments |
Figure 1. It seems like part B of this figure has been omitted. Please add this to the Figure. It could be helpful to at label each image with the pathogen type. Furthermore, clarify that the upper disk is a control (I assume that is the case). |
|
Response to Additional Comment |
Figure 1 was revised. |
|
Additional Comments |
The authors should enhance and revise Table 1. It would be valuable to include the zone of inhibition diameters in addition to the equivalent MICs. Furthermore, the units of the MIC values should be double-checked. For C. albicans and T. rubrum, MIC values for Lys are indicated as “>10 g/l.” Why is the “>” necessary? 10g/l is equivalent to 10mg/ml, so I’m confused why mg/ml are not used here as in the other rows. Please revise. |
|
Response to Additional Comment |
We revised Table 1 and, added the sentence as follow: No inhibition zone diameters of L-lysine at concentration of 1 mg/ml were observed. As it can be observed the antimicrobial efficiency on tested microorganisms was stronger using PLL at concentration 1 mg/ml. |
|
Additional Comments |
The rationale for selecting the concentrations of Lys and PLL should be further explained. It is not clear why these values were chosen for use with the PEF treatments. |
|
Response to Additional Comment |
In the study PLL exhibited very strong antimicrobial activity on all tested microorganisms. However, Lys had no or little effect on the growth of C. albicans and T. rubrum. Considering that the aim of the study was to show the combination of PEF and Lys/PLL, thus the concentrations have been selected to return inactivation rates below 0.25 log by the sole Lys or PLL treatment. |
|
Additional Comments |
Figure 2. The final bar in the plot (25 pulses at 100us duration) is not visible in either plot. Therefore, it is not clear what this value for this condition is. This is understandable for part A, assuming the viability is very low; however, in part B with the log scale, it is unclear whether the value is very close to 0.01 (lowest value on the graph), or if it is below this value. I suggest decreasing the lower bound of the y-axis to show this final bar. |
|
Response to Additional Comment |
Fixed. All graphs have been made to show log CFU and same scale. |
|
Additional Comments |
Figure 5B. A log scale for viability would better show the data for this plot. |
|
Response to Additional Comment |
Fixed. All graphs have been made to show log CFU and same scale. |
|
Additional Comments |
It would be helpful for the authors to include a table summarizing the results from Figures 2-5. In particular, it would be helpful for the authors to state the log reduction of pathogens for each treatment condition. |
|
Response to Additional Comment |
Considering that all the graphs now have been redrawn to show log reduction in CFU and all feature the same scale, now there is no need for a table. |
|
Additional Comments |
The results show Lys and PLL treatment in combination with PEFs can reduce pathogen viability by several log reductions (~1-4). Can you discuss if these reductions are sufficient for disinfection proposes for food/biotechnology applications? It would be good to compare these efficacies with other similar studies (either with Lys/PLL or PEFs with other agents). If these efficacies are not low enough for some applications, could higher voltages or longer pulses potentially increase the efficacy? |
|
Response to Additional Comment |
According to your suggestion, the “Discussion” was modified. |
Reviewer 3 Report
An interesting and innovative work. My comments are below.
Materials and Methods
2.3. Pulsed power setup – in my opinion these methods should be better explained, especially for other readers than physicists. What role does the electroporation cuvette play? What does it mean “electroporation”? Line 101 – I do not understand: “for setup generated single and bursts of 0-30 kV/cm electric field”. The term “energy density” should be explained as well as the interrelations among the number of pulses (2000, 1000 etc.), the duration time of pulses (500 ns, 1 μs etc.) and the energy density. In further paragraphs the term “amplitude of 15 or 30 kV/cm” is used. What does it mean: amplitude = electric field = field intensity? The physical background is poorly explained.
Line 113 – what was the control in this experiment?
Lines 114-123 – it will be better if the lines concerning the disk diffusion test and MIC determination are placed above, between the paragraphs 2.2 and 2.3 as a new separate subsection.
Results
The presented Figures (2 -5) are clear and convincing. However, it is usually established that the disinfection effect is acquired when the decrease in the microorganisms number expressed in the logarithmic scale is per 4 orders of magnitude or more. I see that this result is achieved in the case of the electric field of 30kV/cm, but it will be helpful for assessing the results more precisely, if the Authors calculate the degree in the reduction of microorganisms number in the logarithmic scale additionally.
Lines 207-257 “Discussion” – the results indicate that significant reduction of microorganism number was achieved owing to the pulsed electric field together with PLL. It will be interesting what influence on the quality of food or on human tissues is expected following the practical application of this method. I am especially interested if it will be possible to use the method to kill the microorganisms, mainly fungi, on historic paper, parchment or canvas with paint layers. Could the Authors discuss the possibilities of application in various fields?
Poor English I observed in the following line (I am not a native speaker):
- Line 108 – “80-μl sample”
- Line 110 and line 111 – too many “and”
- Line 117-118 – the whole sentence from “The sterile forceps ...”
- Line 182 – “Duration-wise”
Author Response
Dear Keith Wang (Assistant Editor) and Reviewers of Applied Sciences,
Re: Revision of “Antimicrobial Activity of L-Lysine and Poly-L-Lysine with Pulsed Electric Fields” by Jurgita Svediene, Vitalij Novickij, Rokas Zalneravicius, Vita Raudoniene, Svetlana Markovskaja, Jurij Novickij, Algimantas Paskevicius (applsci-1127909)
Thank you very much for your kind reviews and comments regarding our manuscript (applsci-1127909) entitled above. Now we have carried out revisions according to your comments and hope this will be adequate for the acceptance of this manuscript. Details of corrections according to the comments are as follows;
|
Reviewer #3 |
|
|
Comments |
2.3. Pulsed power setup – in my opinion these methods should be better explained, especially for other readers than physicists. What role does the electroporation cuvette play? What does it mean “electroporation”? Line 101 – I do not understand: “for setup generated single and bursts of 0-30 kV/cm electric field”. The term “energy density” should be explained as well as the interrelations among the number of pulses (2000, 1000 etc.), the duration time of pulses (500 ns, 1 μs etc.) and the energy density. In further paragraphs the term “amplitude of 15 or 30 kV/cm” is used. What does it mean: amplitude = electric field = field intensity? The physical background is poorly explained. |
|
Response to Comment |
We tried to improve the text where possible. Also, we have simplified some sentences. However, all the main parameters are reported according to the standards and rules for reporting electroporation parameters in PEF studies. We don’t have any flexibility in reporting them. Energy density, amplitude, etc. are all fundamental physical parameters measured in SI units. This paper was also submitted to a special issue, which is dedicated to electroporation. |
|
Comments |
Line 113 – what was the control in this experiment? |
|
Response to Comment |
The control samples were the untreated samples. Missing information was added to the text. Also, we have included Lys only and PLL only effects in all graphs to prevent any further confusion. |
|
Comments |
Lines 114-123 – it will be better if the lines concerning the disk diffusion test and MIC determination are placed above, between the paragraphs 2.2 and 2.3 as a new separate subsection. |
|
Response to Comment |
According to your suggestion, we a new separate subsection was created. |
|
Comments |
The presented Figures (2 -5) are clear and convincing. However, it is usually established that the disinfection effect is acquired when the decrease in the microorganism’s number expressed in the logarithmic scale is per 4 orders of magnitude or more. I see that this result is achieved in the case of the electric field of 30kV/cm, but it will be helpful for assessing the results more precisely, if the Authors calculate the degree in the reduction of microorganisms number in the logarithmic scale additionally. |
|
Response to Comment |
All the graphs have been redrawn to be in log scale to allow precise evaluation and comparison. All the scale in all the graphs is now the same. |
|
Comments |
Lines 207-257 “Discussion” – the results indicate that significant reduction of microorganism number was achieved owing to the pulsed electric field together with PLL. It will be interesting what influence on the quality of food or on human tissues is expected following the practical application of this method. I am especially interested if it will be possible to use the method to kill the microorganisms, mainly fungi, on historic paper, parchment or canvas with paint layers. Could the Authors discuss the possibilities of application in various fields? |
|
Response to Comment |
In the context of a food or wound processing the methodology is applicable and at the current state looks promising. However, in the context of historic papers, canvas, etc. the effects of pulsed electric field on ink and paint should be investigated. Also, we afraid that it will have limited applicability in this context since electroporation requires a conductive medium between electrodes (liquid food, tissue, etc.) to be uniform and successful. Therefore, in the case of paper or canvas, we believe that photodynamic therapy or cold-plasma treatment might be a more promising and worthy of investigation. |
|
Comments |
Poor English I observed in the following line (I am not a native speaker): • Line 108 – “80-μl sample” • Line 110 and line 111 – too many “and” • Line 117-118 – the whole sentence from “The sterile forceps ...” • Line 182 – “Duration-wise” |
|
Response to Comment |
We have revised the manuscript. |
Round 2
Reviewer 2 Report
The authors have satisfactorily addressed my comments and significantly improved the manuscript. I recommend accepting the manuscript.